# Recovery-Oriented Cross-Sectoral Network Meetings between Mental Health Hospital Professionals and Community Mental Health Professionals: A Critical Discourse Analysis

**DOI:** 10.3390/ijerph19063217

**Published:** 2022-03-09

**Authors:** Kim Jørgensen, Kate Andreasson, Tonie Rasmussen, Morten Hansen, Bengt Karlsson

**Affiliations:** 1Department of Public Health, Nursing and Health Care, Aarhus University, 8000 Aarhus, Denmark; 2Psychiatric Centre North Zealand, 3400 Hillerød, Denmark; kate.aamund@regionh.dk; 3Department of Social and Health, Center for Quality and Development, 3460 Birkerød, Denmark; tora@rudersdal.dk; 4Psychiatric Outpatient Clinic, Residence Team, 2635 Ishøj, Denmark; aesel82@gmail.com; 5Center of Mental Health and Substance Abuse, Department of Health-, Social- and Welfare Studies, Faculty of Health and Social Sciences, University og Southeastern Norway, 3007 Drammen, Norway; bengt.karlsson@usn.no

**Keywords:** recovery-oriented, cross-sectoral collaboration, intersectoral care, network meeting, healthcare professionals, users, mental health hospitals, community mental healthcare

## Abstract

Aims and objectives: In the medical field, we lack knowledge on how interprofessional collaboration across sectors is carried out. This paper explores how healthcare professionals and users perceive recovery-oriented cross-sectoral discharge network meetings between mental health hospital professionals and community mental health professionals and which discourses manifest themselves within the field of mental healthcare. Method: Ten professionals from a mental health hospital and eight community mental health professionals participated. In addition, five users with experience in mental health services in both sectors participated. Fairclough’s discourse analysis framework was used to explore their experiences. The study was designed following the ethical principles of the Helsinki Declaration and Danish law. Each study participant in the two intersectoral sectors gave their informed consent after verbal and written information was provided. The Consolidated Criteria for Reporting Qualitative Research checklist was used as a guideline to secure accurate and complete reporting of the study). Results: The healthcare professionals in both sectors are governed by steering tools, legislation and a strong biomedical tradition to solve illness-related problems, such that users must be offered treatment and support to achieve self-care as soon as possible. This can be seen as a reflection of, and a driving force in, a change in the wider social practice that Fairclough terms the ‘marketisation of discourse’—a social development in late modernity, whereby market discourse colonises the discursive practices of public institutions. The user of psychiatric and social services experiences a structured system that does not offer the necessary time for deep conversations. Users do not consider recovery as something that is only seen in relation to the efforts of the professionals, as recovery largely takes place independently of professionals. Recovery depends on users’ internal resources and a strong network that can support them on the journey. Conclusion: Healthcare professionals perceive recovery-oriented cross-sectoral discharge network meetings to reflect paternalistic and biomedical discourses. Users want to be seen more as whole persons and did not experience sufficient involvement in the intersectoral care. Relevance to clinical practice: Healthcare professionals need to be supported to seek clarity in the understanding and operationalisation of a recovery-oriented approach, if the agenda is to be truly adopted and strengthened.

## 1. Introduction

In Western society, recovery-oriented cross-sectoral collaboration is a prominent topic in the health sector. Health professionals, users, mental health services and governments aim to strengthen cross-sectoral collaboration [1,2]. Ideally, users should experience coherence in the efforts of services when they are discharged from a mental health hospital for follow-up in the community social psychiatric service. Despite the objective of strengthening cross-sectoral collaboration, there are many barriers to achieving this aim. There is no common understanding of what recovery and cross-sectoral collaboration mean and how they should be implemented in a clinical practice [3]. The plans drawn up in one sector are not used in the next, and users have to start all over again when they change therapists [4]. The professionals do not agree on how to interpret the user’s mental and social challenge with a mental illness. Consequently, there is no common starting point for how professionals should promote users’ recovery processes.

## 2. What Does This Paper Contribute to the Wider Global Clinical Community?

The aim of this study is to explore how healthcare professionals and users perceive recovery-oriented cross-sectoral network meetings and identify which discourses manifest themselves within the field of mental healthcare.

Healthcare in both sectors is governed by steering tools, legislation and a strong biomedical tradition to solve illness-related problems such that users must be offered treatment and support to achieve self-care as soon as possible.

Healthcare professionals want to contribute to a more recovery-oriented discourse in cross-sectoral network meetings. However, the literature reflected that they maintain a treatment structure subjected to a paternalistic, biomedical and legislative discourse.

The professionals lack methods to work in a recovery-oriented way and to ensure coherence in efforts across psychiatric and municipal sectors [5]. Various initiatives are taken in mental health services across the sectors to promote a more recovery-oriented practice and cross-sectoral collaboration, e.g., implement open dialogue [6], flexible assertive community treatment [7], more focus on personal recovery [8,9], and cross-sectoral network meetings [10,11].

Despite the aim in Denmark of focusing on professional efforts on the basis of looking towards recovery, there are still challenges in ensuring a coherent effort directed at recovery across mental health hospitals and community mental healthcare [4,12]. Users do not find that the professionals communicate with each other about plans prepared in one specific sector. They often experience having to start over again every time they switch between treatment locations [1,2,13]. The professionals have therefore implemented cross sectorial network meetings to ensure coherence in efforts across psychiatric and municipal sectors [14]. Despite good intensions, there are several challenges to achieving this ambition. A network meeting includes the user, relatives (optionally), professionals from the mental health hospital who have been involved in users’ treatment and community mental health professionals who will follow up with the user after discharge. The meeting is held immediately before discharge and aims to prepare a recovery-oriented plan for post-discharge. Maximum involvement of the user and any relatives is planned, but in practice, the user does not feel involved and does not always get invited to the meetings [4,5]. Furthermore, professionals lack a common understanding about what recovery means and what it means to strengthen cross-sectoral collaboration [15]. There is a need for the professionals who work together to support users’ recovery processes to collaborate to reach a more unified understanding of the recovery-oriented cross-sectoral network meeting. The last decade has shown the positive outcomes to be gained from users’ knowledge in the development of new initiatives or products that arise jointly between different professional groups and the users of mental health services in hospitals and municipalities [16,17,18]. The medical field lacks knowledge on how interprofessional collaboration is carried out across sectors. This paper explores how healthcare professionals and users perceive recovery-oriented cross-sectoral discharge network meetings between mental health hospital professionals and community mental health professionals and which discourses manifest themselves within the field of mental healthcare. The purpose is to examine the professionals’ and users’ use of spoken language, which reveals what underlies the cross-sectoral collaboration in a mental healthcare context.

## 3. Methods

The research group prepared the research protocol and sought approval for the project from the ethical and legal committee, gained access to the field, and then the researcher and co-researchers discussed and decided on the research design. Professionals and users were recruited as participants in the cooperative process. Participants had experience with mental health services in a mental health hospital and community mental health. Users did not have serious symptoms, such as social anxiety or psychotic delusions. Those who did not want to share their own experience with others in the workshop meetings were excluded [19].

The cooperative inquiry took place in three workshops where the participants discussed their experiences with network meetings and recovery. The purpose was to share experiences and identify challenges in preparing recovery-oriented plans jointly and with maximum involvement of users and any relatives. The research group participated equally as participants and facilitated the process by holding a focus on recovery-oriented network meetings [5,18,20].

All participants sought a common understanding of recovery. There was a need for inspiration to have a theoretical point of view to talk about recovery, and there was agreement that network meetings should be based on a dialogically equal conversation. Everyone’s opinions should be equal, regardless of whether the participant was a user, researcher, or health professionals. Many of the participants were familiar with the recovery CHIME model [3,21] and open dialogue [6]. At the request of the participants, the research group presented this theoretical approach, which was adopted as a framework for network meetings. The workshop meetings were audio-recorded and transcribed and were the subject of subsequent discourse analysis.

### Discourse Analysis as a Social Construction

This study presents itself on a social constructivist basis, assuming that reality is constructed. Reality is not given in advance, nor do humans have a predetermined and inherent nature. This also means that our perception of reality is a product of socio-cultural processes. Reality is only available to us through social interaction processes, and it is through these that we experience and recognise it. Our knowledge does not reflect the world as it is in itself [22,23]. Access to understanding reality takes place through language, social relations and cultural contexts. The concept of discourse originates from the linguistic research tradition, where it refers to the structuring of linguistic statements in systems. The focal point is people’s use of language when we communicate with each other, as well as the power relations that unfold in the communication process. Identity and relationships are therefore created through language. Discourse is a social construction that helps to systematise a common reality based on language [24]. Discourse is a certain way of talking and understanding a part of the world, and language constructs a social practice and is constituted by other social practices; discourse is in a dialectic relationship with other social dimensions; and discourses add to, form, transform and reflect social structures and processes [5,25,26]. The epistemology of this study focuses on the recovery-oriented network meeting as a concept to be deconstructed and examined as discourses. The social realm is seen as being centred around recovery, and network meetings and mental healthcare are conceptualised as social constructions, where the subjects appear as social practices through which texts are constructed, received and interpreted [23,27,28].

## 4. Data Collection

A qualitative research design with purposive sampling was adopted [19]. Ten professionals (nurses, physiotherapists, occupational therapists, physicians and social workers) from a mental health hospital and eight professionals (nurses, educators and social workers) from community mental health participated. In addition, five users with experience with mental health services in both sectors participated. The research group comprised five professionals, three with a Ph.D. from postdoc to professor level, a nurse with an academic degree and an educator with lived experiences in mental health services.

Data were collected from the conversations between all the participation during which they discussed challenges and solutions regarding cross-sectoral network meetings. Three workshops formed the framework for the co-creation approach to the meetings. The participants participated equally and were not subject to the guidance of their leader at workshops. The dialogue between the participants was facilitated by questions. The questions were developed with inspiration from Williams with several validated questions and delimited in relation to the perspective of network meetings in a cross-sectoral context [29,30,31]. The purpose was to discuss opportunities and limitations for a recovery-oriented cross-sectoral collaboration between mental health hospitals and community mental healthcare.

The Consolidated Criteria for Reporting Qualitative Research checklist was used as a guideline to secure accurate and complete reporting in this study (See Appendix A).

## 5. Analytical Strategy

Fairclough’s critical discourse analysis framed the analytical approach to the empirical data. Fairclough’s theory offers a tool for text analysis that makes it possible to gain insight into how discursive processes operate linguistically in texts. Fairclough’s critical discourse analysis emphasises the links between texts and societal and cultural processes and structures through an interdisciplinary perspective, which combines textual and social analysis [22,24,25,32,33]. Discourse reflects how language is expressed and makes sense in a social practice. Further, discourses contribute to the construction of social identities, social relations and systems of knowledge and meaning [25,34]. Language is understood as a communicative event consisting of three dimensions: (i) a spoken or written language text, (ii) a discursive practice that involves the production and interpretation of text and (iii) a social practice [24,25,35]. Those three elements are included in the discourse analysis (Table 1). The textual meaning is understood in relation to the meaning in other texts, and in relation to the social context. Fairclough sees discourse as both constitutive and constituted [24,25,26].

The analysis consists of the three-dimensional framework: as a text, as a discursive practice and as a social practice.

The text analysis: We read each text from the workshops several times to ascertain which discourses arose around recovery-oriented network meetings among the included professionals and users. NVivo 7, a qualitative research software programme developed by QSR International, United States (http://www.qsrinternational.com/products_previous-products_nvivo7.aspx, accessed on 26 January 2022), was used to organize data during the analysis. The total data material consisted of transcribed interviews [36].

The text was analysed line by line and word by word. Two researchers analysed the text separately and reached consensus through an open dialogue. The research group was involved and also reflected on the interpretation of the text to capture the opinions of the participants. The text was analysed line by line and word by word. First, we focused on the grammar and interactional control to analysis how control and asymmetrical relations were negotiated between the participants. Transitivity involved analyses of possibly metaphors, and active or passive use of clauses and nominalisations. Modalities refer to the text content, the way words are pronounced and how often modality features are used [23,29,37].

The discourse practice: Inspired by Fairclough, we focused on the texts’ production by analysing interdiscursivity to view discourse types in the texts, and manifest intertextuality to view how texts are drawn upon in the constitution of the texts. Text distributions refer to intertextual chains to view the discursive texts’ relations, e.g., are they stable, shifting or contested? Coherence refers to the interpretative implications of the intertextual and interdiscursive properties of the texts [23,29,37].

The social practice: The general aim at this level of the analysis was to specify the nature of the social practice of which the discourse practice is a part. This is the basis for explaining why the discourse practice is as it is. Through discussions of the findings of the analysis of the texts and the discourse practices, we reviewed and interpreted the social and hegemonic relations and structures, the orders of discourses, and the ideological and political effects of the discourses, leading to a main interpretation of the social practice of recovery-oriented cross-sectoral network meetings [23,29,37].

Ethical considerations: The management of the mental health hospital and community mental healthcare in the Danish capital gave permission to conduct the study. The management asked potential health professionals and users from a network in the community mental healthcare to participate in the study. Hereafter we contacted the participants and gave verbal and written information about the purpose of the study. All participants gave informed consent. All data from the conversations during the three workshops were transcribed anonymously and a commitment was given that the audio recorded would be subsequently erased after the study is finished. The participants were informed that they could withdraw from participation at any time, with no consequences. The study is approved by the Danish Knowledge for Data Review [J.nr.: P-2019-753]. According to the Helsinki Declaration [38] and Danish law [39], no formal permit from a biomedical ethics committee was required, as the research would not influence the informants, either physically or psychologically.

## 6. Results

In this section, we present our findings, based on the three-dimensional analysis. As mentioned, the three dimensions consist of text analysis, discursive practice and social practice [23,24,25,26,27,28,29,31].

### 6.1. Text Analysis

Vocabulary

Network meetings provided a framework for professionals, users and relatives to create a recovery plan and promote recovery-oriented collaboration and coherence between the two sectors. Network meetings were described as an organisational approach to achieve the aim of recovery-oriented cross-sectoral collaboration. Users did not believe the meetings focused enough on recovery. Instead, they reported the focus was often on practical things and it was questionable as an approach.

Most of the professionals talked about recovery-oriented cross-sectoral network meetings using wording, such as ‘checklist’, ‘laws and paragraphs’, ‘effective approach’, ‘medicine’, ‘treatment’, ‘economic priorities’ and ‘systematic screening’, ‘chime’, ‘open dialog’ and ‘recovery process’. These concepts, on the one hand, relate to a linear and objective language when discussing recovery-oriented cross-sectoral collaboration. On the other hand, the concepts ‘chime’, ‘open dialogue’ and ‘recovery process’ relate to a cross-sectoral network meeting having a personal recovery-oriented focus.

The wording around recovery-oriented cross-sectoral network meetings were guided by all the healthcare professionals’ perspectives, as the following quotations show:
*We tick off a list; where should you get your medicine, where should you sleep, who should do what … So, a lot of coordination, practical**(HPC (*HPC represents health professionals from community mental healthcare.*))*.
*We have §85, which is a long-term process, and we have §82, which is a short-term and much more intensive process, where we intervene quickly to get the users well through (HPC)*.
*It is also about structuring our way of talking about or with users when we do. So, we get to catch the little fine grains that are there. Although the patient may be in crisis**(HPH (*HPH represents health professionals from the mental health hospital.*)).*
*That’s actually how we work here. When you are admitted here, you will be given such a problem-aim list, and then a number of days after, you will be given a treatment plan (HPH)*.


The interpretive perspective and keywords were framed in a professional, structural and organisational language. The professionals want to promote a personal recovery-oriented approach, but this thinking must fit into rational and linear management mechanisms. Narrative language was not used to present the users’ perspective regarding the recovery-oriented cross-sectoral network meeting between a mental health hospital and community mental health.

### 6.2. Grammar of the Texts: Interactional Control

All participants accommodated, listened to and acknowledged all views without regard to whether they came from a user, psychiatrist, nurse, researcher or other professional. All users were highly motivated to share their experiences.

Almost all the healthcare professionals used a kind of transitivity whereby they normatively related to what a recovery-oriented network meeting should be. In the language, passive sentences were used such as:
*Recovery is important (HPH).*
*A lot of listening to users’ wishes (HPC).*
*There is a focus on recovery (HPH).*


In sentence constructions, the agent is deprived of responsibility as sentences appear in passive form and omit the agent.

Several of the healthcare professionals and users used an ambiguous modality. On the one hand, they expressed that they are already doing a lot to work that is recovery-oriented, but at the same time, they recognised many barriers to meeting the aim of recovery-oriented network meetings. Structural conditions make achieving these aims difficult, and variations in professionals’ disciplines between the hospital and municipal sectors mean there is no common language with which to talk about user problems. Users’ problems are interpreted into a biomedical or social science modality, and solutions arise from these two approaches. A legal modality was used for this, as the hospital is obliged to diagnose and treat under the Health Act, while the municipality is subject to several laws, e.g., a service and employment law, where the aim is to initiate activities that can promote users’ abilities to become self-sufficient citizens who can manage independently of professional help.

The healthcare professionals in community mental healthcare used a truth modality with high affinity, such as:
*Physicians are a group who unfortunately do not know much about recovery**(HPH)*.

Or reservation modality, for example:
*We conduct a systematic screening with each patient to assess the risk of extroverted behaviour or the need for coercion and here recovery could be part of the conversations**(HPH)*.
*The recovery plan is not ready for us. I can doubt if you can sometimes get so far in the recovery plan that you get a good plan when you are hospitalised (HPH).*

The users applied a critical objective language, and they believe the professionals, despite limited time, should talk about broader perspectives such as hopes and opportunities, such as:
*The medical environment must talk about these things, it is not so dangerous to talk about the big perspectives (User).*
*Further, users stated that professionals should be open and avoid attitudes like:*
*It can probably not be realised (User).*
*He cannot do it anyway, because then we narrow it down from the start, and then we also close down so that the citizens have the opportunity to actually put into words what kind of thoughts they have cares about the future, hopes and dreams (User).*

## 7. Discourse Practice

All the participants in the workshops were directly involved in constructing and rethinking how cross-sectoral meetings can become more oriented in a recovery-oriented direction. However, the findings reflected a power structure, with professionals primarily focused on living up to rational and structured discourses. This was articulated through specific agendas on legislation, diagnosis, symptoms, financial management tools, treatment and action plans, all of which aim to help as many people as possible who become self-employed, self-sufficient and independent of professional help. For example:
*We use the one that §82 to make the goals for what is to take place. So, the user[s] is helping to put into words what it is for some goals and things we want (HPC).*
*We have this famous template. It is the case that we as a place of treatment, the municipality, the user and the relatives have an agenda. There are not many of the points that are about recovery process (HPC).*

This communicative instance shows that users are objectified to persons with mental health problems who must be fit into a particular legislative and treatment logic. The professionals and users all articulated great affinity for personal recovery as a topic that provides a place to talk about everyday things like hopes and dreams for the future. However, this modality about personal recovery is overshadowed by a system that predominantly measures efficiency.

All the professionals’ and users’ selves and subjectivities were constituted in power relations and various forms of governance, partly discursively in the form of scientific theories, and partly in the form of technologies developed in various institutions. The rational and structured discourses of linearity had a controlling influence on what users should focus on to achieve self-care:
*We focus on where the user should pick up his medicine, where he should sleep, who should do what etc., i.e., a lot of coordination, practical which is not very recovery-supportive, because it does not actually speak into what you actually have need in relation to getting a satisfactory life stacked on its legs (HPC).*

All the users of the psychiatric and social services experienced a structured system that does not have the necessary time for deep conversations. All the users do not consider recovery as something that only relates to the efforts of the professionals, as recovery largely takes place independently of professionals. The contend that recovery depends on one’s internal resources and a strong network that can support one on the journey. As the users described:
*When you are admitted, you feel really, really bad. The biggest challenges are usually sleep, or you feel there is someone after you, maybe you have had some medication failure. And then there is a long way to go before you consider hopes and aims and things like that. I may not think you can make a recovery plan once you are hospitalised (HPC).*
*Recovery takes place predominantly in one’s life outside the professional system and recovery also depends on how strong you are and how strong a network you have (User).*

Existing treatment methods in mental health hospitals are oriented towards a biomedical discourse that is prioritised and legitimised by the professionals as the best bid for a recovery-oriented practice. Most of the professionals in the municipal sector identify themselves as being part of a recovery-oriented culture, but this is again overshadowed by the fact that the persons must be integrated into the service the legislation allows for and with limited time for support. Furthermore, the municipal authorities must make plans so that all persons with mental health challenges, as far as possible, become qualified to be independent of professional help. As the healthcare professionals explained:
*We are part of a system where help, support and activity opportunities are offered according to legislation and the resources are not unlimited either. We help assess how much the user needs (HPC).*
*We must not forget that hospitalisation first and foremost is about clarifying the mental difficulties and thus diagnoses. Even though psychopharmaceuticals are scolded, we must remember that it helps most users to get better (HPC).*

The interdisciplinary discourses in the workshops show a manifest intertextuality that has roots in a predominantly paternalistic discourse. Users experience a lack of involvement in how the treatment is organised. However, more opposing discourses are displayed across the sectors, as the professionals used a categorical modality, which is a characteristic of legal and rational-structured discourse.

Intertextually, recovery is reduced to a rational method on an equal footing with, for example, medication and cognitive behavioural therapy. The interpretative implication is that recoveries undergo a linear and structural discourse where personal recovery is obtained by methods such as cognitive behavioural therapy and where predefined questions guide the user on the path. Recovery is articulated as fighting mental symptoms rather than learning to live with them. The professionals all articulated recovery in a way that distances the process from personal recovery as the user’s perspective is overlooked. The discourse of personal recovery cannot be reduced to a physical object but must be an approach to the ways professionals and users interact”

Personal recovery is not articulated as something individually defined between each user and professional but is instead defined by the professionals’ methods and scientific mindsets. As one healthcare professional described:
*Admissions are often acute. For example, a user comes with medicine failure and then it is about motivating and starting medication as soon as possible. We also get some bits of recovery that we do not pick up. I think it points a lot back to ourselves. To gather the knowledge, we have that points in the direction of recovery (HPC).*

The users all applied manifest intertextuality when they argued for a co-creative and democratic discourse, where common understanding is built up with a view to designing tailor-made courses. The interpretative implication of this intertextuality is a clinical practice that is based on the differences each brings with them, and which must shape the different ways in which the help is organised. Furthermore, personal recovery must be put on the agenda and become part of the language of treatment, for example, in network meetings. Additionally, recovery was presented by the users as a philosophy of believing in humans and their ability to achieve their hopes and dreams of a good and meaningful life. As such, the communicative discourse should promote the possibility of dialogue to gain greater insight and uncover how professionals can contribute to recovery. As one user explains:
*Recovery is not a methodical tool but a thinking that is about believing in each other, showing confidence in users’ judgment and experience-based expert knowledge. It is a trusting and respectful relationship, where equality and encouragement are key elements (User).*

The communicative discourse was transformed to a dialogical discourse where open-ended questions are asked about their own perceptions of problems and needs. Dialogic discourse was interpreted as the embodied experience of the reflective, responsive and more personal practices, in which one’s own inner, as well as outer, “voices” matter and signify meaning. The starting point for the dialogue between professionals and users (and relatives, if present) should be of the human living world and not the diagnoses and symptoms:
*Mental difficulties can be about many things in everyday life. It can be experienced hard and here we can use help for how everyday life can be better and how hopes and dreams can be possible (User).*

Unlike tangible methods such as medication, the dialogical discourse is articulated both as a method with specific attributes—for example, active listeners with open and exploratory question types, pauses, paraphrasing and nonverbal communication—and as a way of being together as humans. Dialogue was presented as emphasising listening and responding to a whole person in a context and building dreams such as getting an education, job, independent housing, strengthening their role in a local environment, achieving stronger empowerment to master challenges, etc. As one healthcare professional stated:
*Dialogue is both a method and an approach, because besides that, it is a very concrete method, it is also a way of thinking and collaborating and respecting other people. It is also a way of thinking about mental difficulties. It’s an approach. And so basically you can say that it is a form of treatment where the citizens’ perspective and resources-and the citizen’s network perspective and resources are in focus (HPC).*

Manifest intertextualities display the dialogical discourse present as an approach to put words into thoughts and feelings and can be a means of greater self-insight. The interpretative implications of the intertextual properties of the texts are that successful communication presupposes that the professional can observe and interpret users’ verbal cues and bodily signals.

## 8. Social Practice

In social practice, the three-dimensional analysis maps the social and cultural relations and structures that constitute the wider context of the discursive practice. Our focus now turns to the broader social practice to which these dimensions belong. Structures of action or language are only maintained by being renewed constantly in action/speech. As such, it is in action/speech that they also fail to be maintained, that they are altered. In other word, structures are reproduced but also transformed in practice. The structures are reproduced or transformed depending on the state of relations or the balance of power between those struggling with a particular sustained domain of practice, such a mental health hospital or community mental healthcare setting. The social practice reproduced a neoliberal governance discourse with a rational-structured political discourse to place an individual responsibility on the user to get well. The healthcare in both sectors is governed by steering tools, legislation and a strong biomedical tradition to solve illness-related problems such that users must be offered treatment and support to achieve self-care as soon as possible. This can be seen as a reflection of, and a driving force in, a change in the wider social practice that Fairclough terms the ‘marketisation of discourse’—a social development in late modernity, whereby market discourse colonises the discursive practices of public institutions [23,24].

Initially, the healthcare professionals communicated their desire to contribute to a more recovery-oriented discourse to cross-sectoral network meetings. However, the texts reflected that they maintained a treatment structure subjected to a paternalistic-, biomedical and legislative discourse. The co-creative and democratic discourse, together with personal recovery, is subject to powerful structural control mechanisms as an approach to assist users in recovery. The discourse of personal recovery and equal dialogue is subject to a social practice that maintains a rational-structured political discourse with the aim of encouraging users to become independent and empowered people who manage without professional help.

## 9. Discussion

Personal recovery is subject to the institutional governing power relations, in which biomedical and social science understandings become prevalent in determining what is possible, and it is articulated as an organisational means to promote cross-sectoral collaboration and coordinate treatment and follow-up social work efforts [39]. Recovery is objectified and becomes a lever to carry out the various efforts that the two sectors find most relevant. At the network meetings, practical matters were prioritised, such as managing medicine, housing and finances, and to fulfil these, attempts are made to ask questions that focus on users’ recovery. The health professionals articulated a motivation to work in a recovery-oriented way, but it is reported in clinical practice that recovery becomes subject to a standardised evidence-based understanding of care and treatment. This approach is based on expert knowledge, which has great legitimacy in today’s healthcare system [40,41]. Therefore, nurses may unwillingly risk neglecting the patient’s own perception of their situation in favour of the actions described, for example, in a standardised procedure. With evidence-based care and treatment, there is thus a risk that nursing will be transformed into a one-sided instrumental practice.

The healthcare professionals articulated their care and treatment approach through linear and objective discourses that were evidence-based and perceived as providing the best basis for healthcare services to create coherence in intersectoral collaboration. Previous studies, focusing on recovery-oriented cross-sectoral network meetings, corroborate our findings of paternalistic steering whereby the professionals chose which efforts and goals were the right ones for the users’ intersectoral journeys [41]. Network meetings are a construction that is informed by the professionals in cooperation with users. However, despite the professionals’ stated desire to be more oriented towards users’ personal recoveries, such meetings are directed by a rational-structured political discourse to place individual responsibility on users to get well.

Dialogical discourse is favoured by the professionals and users of mental health services and is considered as leverage to promote a recovery-oriented approach to network meetings. Dialogical discourse is recognised in other research as an approach to promoting personal recovery [42,43]. This approach is closely tied to the democratisation of discourse whereby professionals and users meet equally and respectfully. Democratisation has been a real force in modern medicine, and even though debate continues around cases where inequality and bigotry are still flagrant, the level and salience of the debate itself indicates that such issues are on the agenda [2,9,11,15]. Democratisation and dialogical discourses are not just methods but also interpersonal ways of meeting each other. The professionals referred to personal recovery as a method that can be obtained using open-ended questions so that the user can put into words their hopes and dreams. However, other research indicates that personal recovery must also be fostered by professionals who believe these hopes are possible and sincerely encourage users to live them out [6,14,39,40,41,44]. Despite that, personal recovery is articulated as being achieved using individual tailor-made help, which contrasts with standardised help that aims to help collectively and with the purpose of achieving self-care. Exaltation of self-care is linked with a neoliberalist discourse that frames the possibilities on offer and aims to meet individual needs, though the users themselves may not have the same goal. The cross-sectoral healthcare system must therefore support a care and treatment approach that enables individuals to govern themselves independently of professional help [2,3,4,9]. The user is responsible for seeking knowledge and skills to cope with their problem and become an active citizen who contributes economically.

## 10. Conclusions

The study findings demonstrate that mental health hospital professionals and community mental health professionals use a dialogical discourse as leverage to promote more coherent transitions between sectors. The participants advocate for the ability of dialogue to foster narrative language and create the opportunity to listen to the user’s perspective and focus on personal recovery. Despite the participants’ willingness to prioritise the dialogical conversation at the network meetings, which aim to ensure relational coordination and coherence between sectors, this dialogical discourse is overshadowed by a rational-structured political discourse. In both sectors, healthcare is governed by steering tools, legislation and a strong biomedical tradition to solve illness-related problems such that users must be offered treatment and support to achieve self-care as soon as possible. This study reveals that health professionals believe that they have a recovery-oriented cross-sectoral approach to collaboration and to ensure coordinate and continuity in the treatment and care, but several challenges appear as they attempt to achieve this aim.

Cross-sectoral network meetings form the framework for conversations in which plans after discharge are drawn up and agreements are entered into between professionals and users. Network meetings must place more focus on the user’s personal recovery through open questions and dialogue where the user’s perspective is involved. Presently, recovery is articulated as a clinical concept that focuses on making users independent of professional help and able to take care of themselves. There is a need for a paradigm shift where the focus is on users’ perspectives, hopes and goals for the future.

The data material on which this study is based comprised a total of 32 health professionals, users and researchers. The participants volunteered after the first author held a meeting to inform them about the project and their potential to participate. The users were recruited from a network who work to develop psychiatry and were very motivated to participate in the project and share their views.

The research process involved an investigative and exploratory approach to an issue that requires not only a thorough and reflective attitude, but also the inclusion of mutual interpretations and views from different positions and perspectives. These mutual interpretations are influences and participation through actions that have created change, knowledge and learning [5,13].

Great emphasis was placed on the participants speaking openly and freely, though the researchers ensured the discussions remained on topic. Participants were overwhelmingly conscientious in focusing on discussing the relevant issues. Although we obtained rich material, a larger-scale study with the inclusion of several psychiatric wards and municipalities could further expand the findings. In the same way, it should be acknowledged that the use of discourse analysis exalts the ability of linguistic meanings to influence social reality, yet it is also influenced by social practice. This method of examining the data material may therefore exclude other relevant angles and views.

Using workshops as a framework for a cross-cutting dialogue was successful because the participants approached the case constructively and offered many considerations about network meetings regarding their experiences of what worked and what could be done to promote a better focus on personal recovery. In terms of the researcher’s role, the main challenge was to avoid being rhetorically controlling in a particular direction but to instead to allow the participants speak and discuss based on what they found meaningful.

## Figures and Tables

**Table 1 ijerph-19-03217-t001:** The Fairclough inspired main analytical question [23].

Text Analysis	Vocabulary		How are meanings worded? What interpretative perspective underlies this wording? What are the ‘keywords’?
	Grammar	Interactional control	To what extent is control negotiated as a joint accomplishment of participants, and to what extent is it asymmetrically exercised by one participant?
		Transitivity	What process types are most used, and what factors may account for this? Is grammatical metaphor a significant feature? Are passive clauses or nominalisations frequent, and if so, what functions do they serve?
		Modality	What sort of modalities are most frequent? Are the modalities predominately subjective or objective? And what modality features are most used?
Discourse Practice	Text production	Interdiscursivity	What discourse types are drawn upon in the texts, and how?
		Manifest intertextuality	What other texts are drawn upon in the constitution of the texts, and how?
	Text distribution	Intertextual chains	What sorts of transformation does this (type of) discourse sample undergo; are they stable, shifting or contested?
	Text consumption	Coherence	What are the interpretative implications of the intertextual and interdiscursive properties of the texts?
Social Practice	What is the nature of the social practice of which the discourse practice is a part—why is the discourse practice as it is?		

## Data Availability

No new data were created or analysed in this study. Data sharing is not applicable to this article.

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
