# Peer review of "Recovery-Oriented Cross-Sectoral Network Meetings between Mental Health Hospital Professionals and Community Mental Health Professionals: A Critical Discourse Analysis"

_ijerph, 2022, doi:10.3390/ijerph19063217_

Round 1

Reviewer 1 Report

Thank you for the opportunity to review this interesting manuscript. I have a few minor comments for the authors to consider:

  • Abstract:
    • Aims and objectives: Whilst the authors have stated clearly the people (health professionals and users), factor investigated (network meetings), it would be good to also state the main outcome measure(s )of interest clearly. E.g. (lack of) clarify around the process, etc.
    • Results: Please include the actual results, e.g. n or % the participants who “want to contribute to a more recovery-oriented discourse in cross-sectoral network meetings”
  • Introduction:
    • Please highlight the reason the research question is a problem in Denmark and not elsewhere. The authors may also refer to the principles and/or effective practices applied in other mental health setting, and how those theories can inform the direction of this investigation.
  • Methods:
    • Please cite references to support the method used in this study, i.e. those stated in the first 2 paragraphs of the methods section.
  • Data collection:
    • 1st paragraph: Please add a sentence to clarify if the recruitment was convenience sampling or snowballing technique. The authors then need to discuss impact of such recruitment method on the generalisability of the findings.
    • 2nd paragraph: Please provide details around the conversations. E.g. Were they formal or informal conversations? Could there be any power issue between the interviewer(s) and the participants that affected how they expressed their opinions? Please also state the specific lead questions (and other probing questions as Appendix or Supplementary File). Were the questions assessed for its face validity, content validity, internal consistency, etc through a panel of expert on the face validity and content validity? If not please add this as a limitation of the present study.
  • Text analysis:
    • Please clarify if the analysis was done by (e.g. two) independent research personnels, and how consensus were decided (e.g. through a third personnel) to capture and reflect the opinions of the participants.
  • Results:
    • When citing the quotes or main messages from the participants, the authors may italicised the text or use quotation marks (“ ”) to help readers gauge the key finding.
    • It would be good for the authors to note the number (or %) of participants who shared similar quote. This could be done by adding “(n=x,)” or “(x% of the participants)” (the authors may also specify that it is x% of the users, or x% of the health professionals, if preferred) to the most commonly identified quotes.
  • Discussions:
    • Please add comments around the recruitment method (which can affect the representativeness of the findings) and validity and reliability of the questions (which can affect measurement bias) as additional limitations of this study.

Author Response

Please see add document 

Reviewer 2 Report

The text fits the scope of the International Journal of Environmental Research and Public Health as a whole and of the section (Mental Health) and special issue (Promoting Recovery in Mental Health - Perspectives and Experiences of Professionals and Service Users: 2nd Edition), in particular. The theme is somewhat interesting and up-to-date and is well addressed by the authors. The structure of the text is adequate and is what one expects from a scientific work. The abstract is not clear and should be improved (for instance, it repeats “The healthcare in both sectors is governed by steering tools, legislation and a strong bio-medical tradition to solve illness-related problems such that users must be offered treatment and support to achieve self-care as soon as possible”, and that repetition is absurd; also, nurses are mentioned without context). The keywords and introduction are informative. The overview of the relevant literature is up to date, a plus of the paper. The text is interspaced with quality international journal references, denoting the authors have some proficiency about the subject. Methodologically all procedures are correct.

The topic of the paper is original and adds, in an incremental fashion, to the subject area published content. The text lacks clarity and, at spaces, in not easy to read. One can perceive that the authors are not English natives. The problem starts with the title and is repeated throughout the text, with the use of the expression: “between mental health hospital and community mental health”. This expression is not precise. It should be: “between mental health hospital professionals and community mental health professionals”. I emphasize that this change should be made across the paper. Finally, it is important to note that the conclusions are, in my perspective, consistent with the evidence and arguments presented and that the authors address the main question posed at the beginning of the text, but some of the more relevant literature should be revisited in this section. It would also help if the limitation and relevance to clinical practice section to be merged with the conclusion section.

Furthermore, the “Appendix S1” is nowhere to be found.

Author Response

please see add document 

Round 2

Reviewer 2 Report

Given that the authors made most of the suggested improvements and taking into account the assessment previously made during the first submission, I am satisfied with the new version and consider that the article is now of sufficient quality to be published by the International Journal of Environmental Research and Public Health as a whole in the section (Mental Health) and in the special issue (Promoting Recovery in Mental Health - Perspectives and Experiences of Professionals and Service Users: 2nd Edition), in particular.